# Use of technology to prevent, detect, manage and control hypertension in sub-Saharan Africa: a systematic review

Katy Stokes [ID],[1] Busola Oronti,[1] Francesco P Cappuccio [ID],[2] Leandro Pecchia[1]

[1]School of Engineering, University of Warwick, Coventry, UK
[2]Division of Health Sciences, University of Warwick, Warwick Medical School, Coventry, UK

**Correspondence to**
Katy Stokes;
katy.stokes@warwick.ac.uk

## ABSTRACT

**Objective** To identify and assess the use of technologies, including mobile health technology, internet of things (IoT) devices and artificial intelligence (AI) in hypertension healthcare in sub-Saharan Africa (SSA).

**Design** Systematic review.

**Data sources** Medline, Embase, Scopus and Web of Science.

**Eligibility criteria** Studies addressing outcomes related to the use of technologies for hypertension healthcare (all points in the healthcare cascade) in SSA.

**Methods** Databases were searched from inception to 2 August 2021. Screening, data extraction and risk of bias assessment were done in duplicate. Data were extracted on study design, setting, technology(s) employed and outcomes. Blood pressure (BP) reduction due to intervention was extracted from a subset of randomised controlled trials. Methodological quality was assessed using the Mixed Methods Appraisal Tool.

**Results** 1717 hits were retrieved, 1206 deduplicated studies were screened and 67 full texts were assessed for eligibility. 22 studies were included, all reported on clinical investigations. Two studies were observational, and 20 evaluated technology-based interventions. Outcomes included BP reduction/control, treatment adherence, retention in care, awareness/knowledge of hypertension and completeness of medical records. All studies used mobile technology, three linked with IoT devices. Short Message Service (SMS) was the most popular method of targeting patients (n=6). Moderate BP reduction was achieved in three randomised controlled trials. Patients and healthcare providers reported positive perceptions towards the technologies. No studies using AI were identified.

**Conclusions** There are a range of successful applications of key enabling technologies in SSA, including BP reduction, increased health knowledge and treatment adherence following targeted mobile technology interventions. There is evidence to support use of mobile technology for hypertension management in SSA. However, current application of technologies is highly heterogeneous and key barriers exist, limiting efficacy and uptake in SSA. More research is needed, addressing objective measures such as BP reduction in robust randomised studies.

**PROSPERO registration number** CRD42020223043.

## Strengths and limitations of this study

► This is the first systematic review for use of technologies for hypertension healthcare in sub-Saharan Africa, providing a comprehensive review of the state of the art.
► Heterogeneity of included studies was too high for meta-analysis; therefore, results are reported narratively.
► Grey literature was not searched.
► The search was limited to studies published in English language.

middle-income countries (LMICs).[1 2] Hypertension (high blood pressure (BP)) is considered by the WHO to be the leading risk factor for developing CVD[1] and by the Pan-African Society of Cardiology as the highest priority area for reducing heart disease and stroke in Africa.[3] In sub-Saharan Africa (SSA), the prevalence of hypertension is high, especially in younger subjects, estimated at 46% of the adult population, in contrast with 35% in high-income countries.[3] Reasons proposed for this include urbanisation, increase in life expectancy and lifestyle factors such as poor diet, physical inactivity and smoking.[4] A meta-analysis performed by Atakite *et al* reports that of those with hypertension in SSA, only 27% were aware of their condition, 18% were receiving treatment and 7% had controlled BP.[5]

Low numbers of trained healthcare providers combined with a lack of evidence-based guidance and a high cost in accessing healthcare services for patients in SSA are major challenges.[6] Cost-effective technologies will likely play a critical role to overcome such barriers, through decision support tools[7 8]; dissemination of health information including education and treatment reminders[9 10] and collection and storage of medical data.[11 12] Indeed, the value of information and communication technologies (ICT) to health services has been recognised

## INTRODUCTION

Cardiovascular disease (CVD) remains the most common cause of death due to non-communicable disease (NCD) worldwide, with 78% of deaths occurring in low and

by the WHO for over 10 years.[13] eHealth is defined by the WHO as 'the cost-effective and secure use of ICT in support of health and health-related fields, including healthcare services, health surveillance, health literacy, and health education, knowledge and research'.[14] In this way, eHealth can be delivered through several key enabling technologies (KETs): mobile phone technology, artificial intelligence (AI) and the internet of things (IoT). Mobile phone use is high in SSA, 45% of the population subscribe to mobile services and this use is projected to increase.[15] Research interests into use of mobile phones for healthcare purposes in SSA primarily concern either infectious disease or maternal and child health,[16–18] but attention to NCD is growing.[19 20] AI has many possible definitions, in essence describing a motivation to replicate and automate human cognitive functions, having a myriad of healthcare applications,[21] which have been exploited in high-income countries. Although research in LMICs is relatively limited,[22] drivers such as high disease burden, few qualified healthcare workers and increasing phone and internet connection may drive a rapid advance in AI for healthcare in LMICs.[23] Wahl *et al* describe uses of AI in resource-poor settings, including expert systems assisting or compensating for a lack of personnel, health monitoring using natural language processing and signal processing for diagnostics.[22]

Successful application of the aforementioned technologies for tackling hypertension relies on a strong evidence base in design and implementation. In this way, this work seeks to systematically review the literature regarding the application of mobile phone technologies, AI and the IoT as KETs for healthcare provision for hypertension in SSA. The primary objective is to determine how and which KETs have been used, secondary concerns include study design, setting, quality and findings of outcomes relating to hypertension.

## METHODS
### Search strategy and selection criteria
The systematic review of KETs for healthcare provision for hypertension in SSA followed Preferred Reporting Items for Systematic Reviews and Meta-Analyses 2020 guidelines.[24] We searched Embase, MEDLINE, Web of Science and Scopus electronic databases for studies published in English language only. The search was run from database inception to 23 November 2020 and updated on 2 August 2021.

Search terms covered hypertension (eg, "hypertension", "high blood pressure"), Artificial Intelligence (eg, "AI", "machine learning"), mobile phones (eg, "mobile phon*", "mobile"), internet of things (eg, "internet of things", "iot"), point of healthcare cascade (eg, "prevention", "screening") and countries of SSA (full strings in online supplemental S1).

Studies seeking to assess the application of KETs in SSA, for any point in the healthcare cascade for hypertension, were considered for inclusion. There were no restrictions set on study methodology in terms of participant recruitment, age or comorbidity. Studies were required to be conducted using populations from SSA countries, or from a pool of countries including at least one SSA country. For inclusion, studies must have provided an evaluation of the use of KETs for any aspect of healthcare for hypertension or used AI models for predicting or detecting significant events. Studies that focused only on prevalence or risk factors, that is, used statistical methods but did not develop AI-based predictive models, which were considered out of the scope of this review. The study protocol was registered with PROSPERO: International Prospective Register of Systematic Reviews and is found at: http://www.crd.york.ac.uk/prospero

### Data analysis
Screening was completed independently by two authors (KS and BO). A reference search was conducted on any relevant review articles retrieved. For included studies, data were abstracted to a shared Microsoft Excel document, covering study design, study setting and population (age, demographics, comorbidity), details of KET used, study outcomes, controls/comparators (where applicable), target user (where applicable) and indications of acceptability to user (if provided). For randomised controlled trials, we sought to extract mean baseline and end point BP measurements (in mm Hg), with SD, for the intervention and control groups. If SD was not reported, it was calculated using the CI, as described in the Cochrane Handbook for Systematic Reviews of Interventions.[25] In the event that participants were lost to follow-up, the final number of participants who completed the study protocol was extracted. As heterogeneity among studies was high (no two studies evaluated the same intervention), we used a random-effects model to establish the effect of KET-based interventions on systolic BP reduction. We did the analysis in Open Meta-Analyst,[26] an open-source, cross-platform software for meta-analysis.

Methodological quality was assessed independently by two review authors (KS and BO) using the 2018 Mixed Methods Appraisal Tool[27] for assessing the quality of either quantitative, qualitative or mixed methods studies. Criteria were graded as 'unmet', 'met' or 'can't tell'. For mixed methods studies, provided most criteria were met (three or more out of five) for each component, the components were considered to have adhered to their respective quality criteria (criterion 5.5).

### Role of funding source
The funder of the study had no role in study design, data collection, data analysis, data interpretation or writing of the report.

### Patient and public involvement
Patients and the public were not involved in this research.

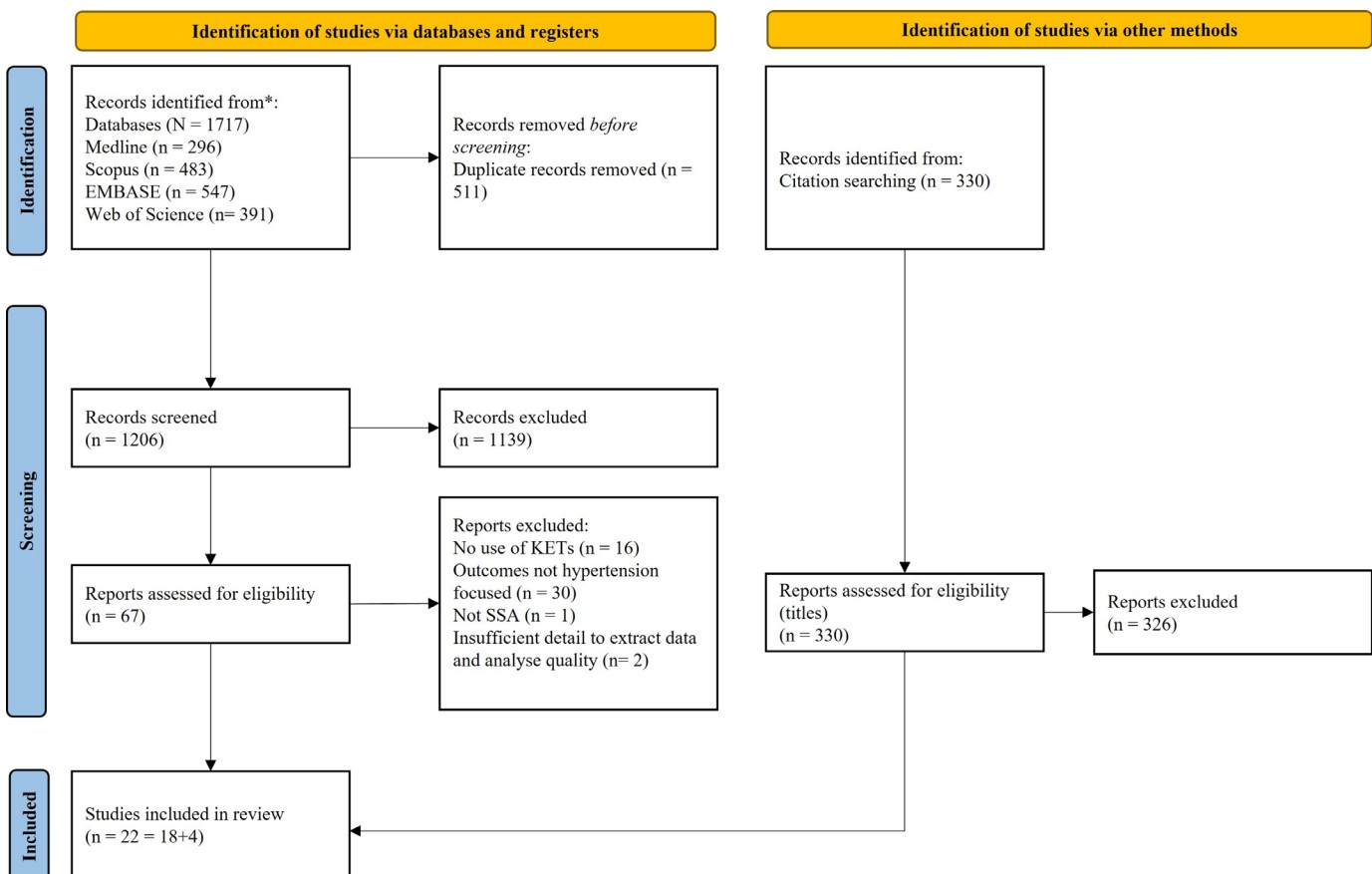

**Figure 1** Study selection. KETs, key enabling technologies.

## RESULTS

Searching MEDLINE, Scopus, Embase and Web of Science yielded 1717 titles. After duplicate removal, 1206 titles/abstracts were screened, with 1139 excluded. Of the remaining 67 full texts, 18 studies were found to meet the inclusion criteria (figure 1). A further four studies were identified through a linear search of the bibliographies of relevant reviews identified during the initial screening. Table 1 contains a summary of the characteristics of the included studies.

Study design, participants, location and aims were highly heterogenous (table 1). The distribution of studies by country is shown in figure 2. All included articles were reported on clinical investigations.[28–49] Two were observational studies[30 46] and 20 concerned evaluation of KET-based interventions.[28 29 31–45 47–49] Both observational studies used mixed methods to explore either current use of technology for hypertension management or hypertension prevalence, understanding and awareness. Several of the interventional studies fell within the same larger study, granting 14 unique experimental studies. Separate articles within these studies reported on different aspects such as impact of intervention, feasibility and perceptions. Eight studies were randomised controlled trials.[28 31 33 34 39–43 45 47] Quantitative primary outcomes were BP reduction and BP control. Other outcomes included treatment adherence, retention in care, awareness/knowledge of hypertension and completeness of medical records. Length

of exposure was highly heterogeneous, ranging from 17 weeks to 2 years.

Quality was highest for qualitative and quantitative descriptive methods and relatively low for randomised or non-randomised studies (online supplemental table S2). Eleven of 13 studies comprising qualitative components satisfied all criteria,[29 31–35 38 43 46 47 49] whereas insufficient reporting of results in two studies meant it was not possible to determine if findings had been adequately derived or substantiated from the data.[37 45] In terms of randomised controlled trials, four met all quality criteria,[33 39 41 42] three did not use appropriate randomisation methods[28 34 40] and three did not report complete outcome data.[34 40 45] Notably, Barsky *et al* did not report sufficient information to judge four out of five criteria[45] and Vedanthan *et al* failed to meet any criteria.[40] Lack of complete outcome data was also an issue in four out of six non-randomised studies.[35 38 44 48] Three of four quantitative descriptive studies met all quality criteria[30 37 42] with one study subject to voluntary selection bias, which was discussed by the authors.[49]

All studies employed adult populations, with varying age requirements (table 1). Five studies used subjects or data collected in a predominantly urban setting,[31 34–36 38] six in rural locations,[28 32 37 40 45 46] two in both[30 41] and one study did not provide specific location or population demographic details.[29] The majority of the experimental research recruited subjects with elevated

**Table 1** Characteristics of included studies

| | Study location | Population (age (SD)) | Duration | Sample size |
|---|---|---|---|---|
| Kingue et al[28] | Yaounde, capital city of Cameroon and rural health districts (within 50–250 km), Telemedicine centre based at Yaounde General Hospital | Age >15, with hypertension not at target level (SBP (or DBP) ≥140 (90) mm Hg or ≥130 (80) mm Hg (for those with diabetes or nephropathy). (Control: 57.6 (12.1), Intervention: 59.9 (10.4)) | 24 weeks | 30 healthcare centres (10 intervention, 20 control). Total: 268 participants (A: Intervention n=165, B: Control n=103) |
| Ola-Olorun et al[29] | Nigeria, (Outpatient clinic Obafemi Awolowo University Teaching Hospital) | Long-term hypertension patients | | Total: 187 participants (exposed to SMS, n=111) |
| Joubert et al[30] | Botswana (suburb) | Adults (>18) (39 (16)) | NA | Total: 92 participants |
| Leon et al[31] (STAR) | South Africa, Cape Town, Primary care facility of a large public sector clinic, | A diverse sample of population of Bobrow et al 2016[33] | NA | 22 trial participants took part in two focus groups, 15 individual in-depth interviews |
| Vedanthan et al[32] | Kenya (rural) | Nurses, clinical officer | NA | Total: 13 participants (12 nurses, 1 clinical officer) |
| Bobrow et al[33] (STAR) | As above, Leon et al 2015[31] | Adults (≥21) with access and ability to use a mobile phone for SMS; diagnosed with hypertension; prescribed blood pressure lowering medication; and with SBP <220 mm Hg and <120 mm Hg at enrolment. (usual care: 54.7 (11.6), information only: 53.9 (11.2), interactive: 54.2 (11.6)) | 12 months | Total: 1372 participants (A: information-only SMS text-messages n=457, B: interactive SMS text-messages n=458, C: usual care n=457) |
| Hacking et al[34] | South Africa: Gugulethu township of Cape Town (densely populated, poor urban settlement) | Patients of hypertension clinic. (52.83 (11.62)) | 17 weeks | Total: 223 participants, (Intervention n=109, Control n=114) |
| Haricharan et al[35] | South Africa, Cape Town | Convenience sample | 28 weeks | Total: 41 participants |
| Kleczka et al[36] | Kenya, Nairobi Health Centre | Patient charts classified with hypertension | 6 months | Total: 70 patients' charts (291 clinical encounters for HTN across 49 patients (149 pre-intervention and 142 post-intervention)) |
| Mannik et al[37] (AFYACHAT) | Kenya (rural), Two rural primary health clinics: Isiolo District, Marakwet District | Adults (>40 years) (50 (43–60)) | 22 months | Total: 2865 participants |
| Nelissen et al[38] | Nigeria (Lagos) | Hypertensive adults (54.9 (11.9)) | 6–8 months | Total: 336 participants |
| Sarfo et al[39] (PINGS) | Ghana, Outpatient Neurology clinic (Komfo Anokye Teaching Hospital KATH) | Adults >18, recently confirmed stroke (<1 month) by CT, with uncontrolled hypertension (SBP ≥140 mm Hg) (55 (13)) | 3 months/9 months | Total: 60 participants (Intervention n=30, Control n=30) |
| Vedanthan et al[40] | Western Kenya: rural healthcare facilities in Kosirai and Turbo divisions | Adults, with elevated BP (SBP ≥140 or DBP≥90). (60.8 (14.2)) | 15 months | Total: 1460 participants (A: usual care n=491, B: paper-based n=500, C: smartphone n=469) |

Continued

| Table 1 | Continued | | | |
|---------|-----------|--|--|--|
| | Study location | Population (age (SD)) | Duration | Sample size |
| Owolabi et al[41] (THRIVES) | Nigeria, A range of facilities chosen to represent the diverse South-western population and hospital types | Adults ≥18 with access to a mobile phone, recently discharged from hospital following a stroke. (57.2 (SD 11.7)) | 12 months | Total: 400 participants (Intervention n=200, Control n=200) |
| Sarfo et al[42] (PINGS) | As above, Sarfo et al[39] | Adults >18, recently confirmed stroke (<1 month) by CT, with uncontrolled hypertension (SBP ≥140 mm Hg) | 9 months | Total: 60 participants (Intervention n=30, Control n=30) |
| Nichols et al[43] (PINGS) | As above, Sarfo et al[39] | | | 24 patients, 8 caregivers, 7 research team |
| Cremers et al[44] | As above, Nelissen et al[38] | As above, Nelissen et al[38] | NA | In-depth interviews total: 30 patients (9 community pharmacists, 6 cardiologists) Structured interviews total: 328 patients |
| Barsky et al[45] | Tanzania (rural) | Adults (≥18) with uncontrolled hypertension. Either own mobile or be willing to take one | 10 months | Total: 130 participants |
| Oduor et al[46] | Kenya (rural) | Adults with HIV and hypertension | NA | Total: 36 participants (27 medical practitioners, 9 patients) |
| Adler et al[47] | Ghana, Lower Manya-Krobo District (84% urban population) | Patients, nurses, clinicians, physician's assistant, pharmacist | | Total: 55 participants (15 patients, 7 nurses, 1 clinician, 1 physician assistant, 1 pharmacist) |
| Vedanthan et al[48] | As above, Vedanthan et al[32] | Adults (>35) Confirmed diagnosis of hypertension (61 (13.2)) | 3 months | Total: 1051 participants (180 under care of nurse, 871 under care of clinical officer) |
| Aw et al[49] (AFYACHAT) | As above, Mannik et al[37] | Adults (>40 years) (50 (43–59)) | 5–8 months | Total: 1650 participants |

DBP, diastolic blood pressure; SBP, sytolic blood pressure.

BP[40] or confirmed hypertension/prescribed antihypertensives,[28 29 31 32 34 36 38 39 47] with the aim of improving hypertension control, treatment adherence or health knowledge, otherwise, the aim of the study was to test a health tool for identifying hypertension or general CVD risk factors within a certain population.[37] In terms of comorbidity or other conditions, one study focused on patients having recently suffered stroke (with or without hypertension),[41] one focused on diabetic patients[28] one concerned HIV-positive hypertensive subjects[46] and one employed a convenience sample of a Deaf community.[35] In three cases, participants were required to have access to a mobile phone for inclusion in the study.[31 41 47]

Table 2 describes the different applications of technologies and their frequency of use. All studies used mobile phone technology, including Short Message Service (SMS), phone calls and mobile applications (apps), either alone or in combination. Odour et al evaluated the general use and perceptions of medical practitioners and patients towards technology, in particular, mobile technology, in dealing with hypertension and HIV. IoT devices employed were automatic BP monitors and their use was also facilitated by mobile phones.

SMS messages were mostly targeted to patients, for health knowledge improvement,[34 35] motivation/improved treatment adherence[29 39 41] or both.[31] Content included reminders for taking medication or attending clinics/appointments, educational information covering general healthy lifestyle suggestions (eg, eating habits, exercise) or hypertension information (eg, symptoms, further health consequences, medication information). SMS was also used in combination with other elements in broader interventions to facilitate decision support for healthcare providers (eg, through direct feedback of risk

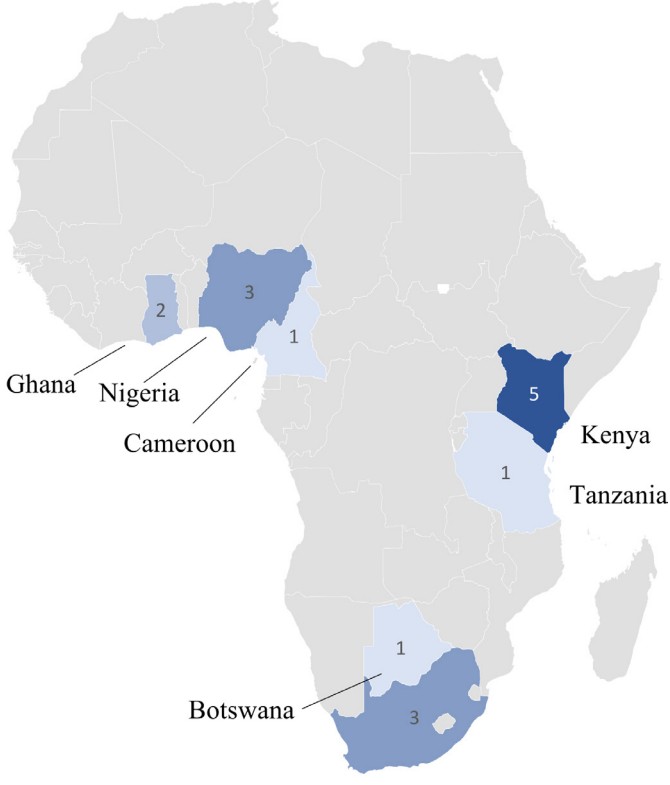

**Figure 2** Distribution of studies across sub-Saharan Africa. Countries are coloured based on the number of studies conducted (darker indicates more studies) and annotated with frequency (where a large study had several associated publications, the location is reported once).

stratification)[28 37] or to provide an interactive connection between patients and pharmacists[29] or specialist telehealth connection (eg, direct feedback from cardiologist to pharmacists).[28 44]

Apps were targeted to healthcare providers, most frequently community health workers. Three studies used apps designed to mediate BP reading collection and dissemination of results for risk assessment and follow-up.[42 45 49] The DREAM-GLOBAL app received BP readings from an automatic monitor via Bluetooth (UA-767 Plus BT), relayed the readings to a remote central server, which then calculated the average reading and transmitted the patients' results to their primary healthcare provider by fax, and to the patient themselves by SMS.[45] For high BP readings, the patient was notified to seek advice with their healthcare provider. Similarly, in the Phone-based Intervention under Nurse Guidance after Stroke (PINGS) study, the same Bluetooth BP monitor was linked with an app for monitoring and reporting measurements as well as medication intake. Participants monitored their own BP, following training from a study nurse. Levels of medication intake were monitored, and tailored motivational SMS was delivered to participants based on these results.[39] The AFYACHAT mobile app functioned in a similar way, BP readings were entered by the CHW, along with other patient data,

the app then provided an algorithmic risk stratification (based on WHO's prevention of CVD: Pocket Guidelines for Assessment and Management of Cardiovascular Risk, 2007) via SMS.[37 49] The Decision-Support and Integrated Record-Keeping (DESIRE) tool, an app designed for nurses to use on tablets, also provided clinical decision support, here through the AMPATH hypertension management algorithm, which is based on WHO clinical algorithms.[32] The DESIRE tool included functions for data entry and validation, decision support, alerts and reminders, and viewing historical data on. Other authors discussed the feasibility of mobile technology-facilitated screening for hypertension. Joubert et al collected survey data using a tablet computer to collect and relay patient information to a central database.[30] Another focus of research was digitisation and storage of patient data from previous paper-based systems. Kleczkaa et al described use of rubber stamp templates containing checklists of clinical practice guidelines; smartphone cameras were used to take images of these templates, which were then manually synched to a cloud-based database, with plans for further automation.[36] A cloud-based health record system was also used as part of the ComHIP hypertension improvement project, which facilitated delivery of SMS and aimed to allow all levels of health providers access to patients' records.

Concerning outcomes relating either to reduction of BP or improved BP control, the majority of experimental studies reported that their interventions resulted in improvements.[28 33 38 40 42] However, analysis by Nelissen et al[38] found that the mobile health (mHealth) app element of their intervention was not associated with the observed BP improvements, based on duration of patient activity measured by the app.[38] Four studies detected no difference[41] or statistically insignificant changes[33 39 40] between control and intervention groups. Vedanthan et al observed significant reduction in both systolic and diastolic BP regardless of whether their tablet-based decision support tool was used by nurses or clinical officers but did not have a control group. Heterogeneity in both outcomes investigated and reported prevented quantitative comparison. Results from three randomised controlled trials which reported baseline and endpoint values for systolic BP are presented in figure 3. It should be emphasised that these trials differed greatly in their intervention plan, study design and location (see table 1), and, therefore, it was not appropriate to report an overall effect. Bobrow et al and Owolabi et al met all quality criteria, however, Vedanthan et al[48] failed to meet any, with authors describing difficulties in data collection and high levels of missing data. Some authors stated it had not been feasible to power studies to detect significant BP reduction, for example, the 3-month interim results of the PINGS trial did not find significant BP reduction due to the intervention until after 9 months, when the proportion of participants with controlled BP became significantly higher in the intervention arm (46.7% vs 40%).[39 42] In some cases, authors noted that effects varied between subjects

**Table 2** Summary of KETs used in study pool

| | SMS (11) | Smartphone/ tablet with app (8) | Mobile/smartphone/ tablet without app (6) | IoT (3) | Web-based data storage and tools (7) | Description of technology | User |
|---|---|---|---|---|---|---|---|
| StAR: SMS-text Adherence Support[31 33] *† | + | | + | | | SMS sent to patients to elicit behavioural changes, focusing on providing educational and motivational messages about hypertension and its treatment | Patient |
| LARK: Linkage And Retention to hypertension care in rural Kenya, Vedanthan et al[40] *† | | + | | | + | Smartphone linked to electronic health record: Provides CHW with automatically updated list of patients requiring follow-up and real-time decision support using clinically approved care algorithms | CHW |
| Kingue et al[28] *† | + | | + | | | Mobile phone communication: Links with telemedicine centre via SMS, voicemail and phone calls. Real-time feedback to aid decision making. | Healthcare provider |
| Owolabi et al[41] *† | + | | | | + | SMS messages for appointment reminders and self-management support. | Patient and care provider |
| Hacking et al[34] *† | + | | | | | SMS messages containing information on hypertension and healthy lifestyle suggestions. | Patient |
| PINGS: Phone-based Intervention under Nurse Guidance after Stroke[39 42 43] *† | + | + | | + | + | BP reading device, connects via blue tooth and smart phone given to patients, stores and reports BP measurements and medication intake. Also, motivational SMS based on adherence to medication. | Patient |
| ComHIP: Community-based Hypertension Management Project[47] *† | + | | + | + | + | Telemedicine consultation by CVD nurse with physician in order to refer serious hypertension on, ICT messages for healthy lifestyles, treatment adherence support and treatment refill reminders, Cloud-based EMR system linked with SMS/voice messaging for treatment adherence, reminders and health messaging, digital sphygmomanometer | Patient and care provider |
| DESIRE: Decision-Support and Integrated Record-keeping[32 38 48] * | | + | | | | Tablet-based Decision Support and Integrated Record-keeping | Nurses/clinical officers |
| Pharmacy task shift[44] * | | + | | | | mHealth mobile application to facilitate communication between pharmacists and cardiologists | Pharmacists and remote cardiologists |

Continued

**Table 2** Continued

| | SMS (11) | Smartphone/ tablet with app (8) | Mobile/smartphone/ tablet without app (6) | IoT (3) | Web-based data storage and tools (7) | Description of technology | User |
|---|---|---|---|---|---|---|---|
| AFYACHAT: health chat[37 49]* | + | + | | | + | mHealth mobile application for data collection including an algorithmic risk stratification based on WHO guidelines | CHWs |
| Ola-Olorun et al[29]* | + | | | | | SMS messaging to connect patient to pharmacist and also to deliver reminders for medication and clinic appointments to patients | Patient and pharmacist |
| Kleczka et al[36]* | | | + | | + | Digital data extraction and management, including guidelines for specific diseases to be stamped, filled and digitised using mobile phones | Clinical staff |
| Haricharan et al[35]* | + | | | | | SMS containing information on hypertension (eg, symptoms, consequences) and tips for healthy living (eg, eating habits, exercise) | Patient (public, deaf) |
| Barsky et al[45]*† | + | + | | + | + | Bluetooth-enabled blood pressure monitor, linked to a mobile phone with DREAM-GLOBAL app to collect readings. Central server assessed readings as normal or high. SMS directed to patient to prompt seeking healthcare | CHW, patient |
| Oduor et al[46] | + | + | + | | | Any reported by participants | Patients and care providers |
| Joubert et al[30] | + | + | + | | | Tablet computer used to collect survey data and transmit via tele-contact | Clinical staff |

*Interventional studies.
†Randomised control trials.
BP, blood pressure; CHW, community health worker; CVD, cardiovascular disease; EMR, electronic medical records; ICT, information and communication technologies; KETs, key enabling technologies.

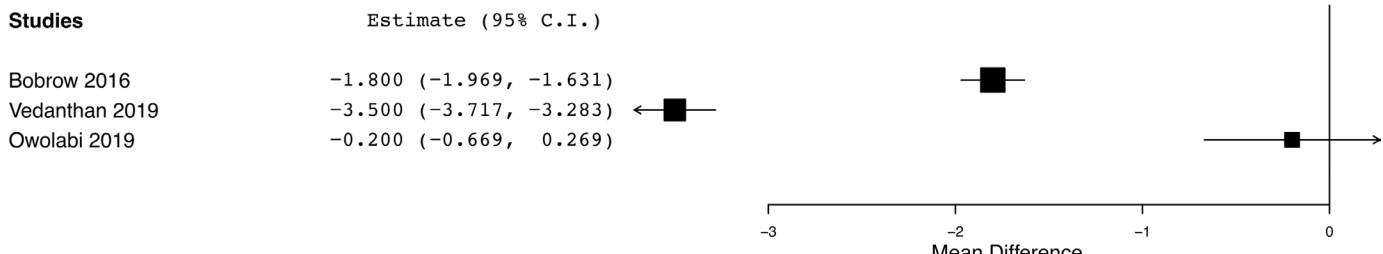

**Figure 3** Change in mean systolic blood pressure (mm Hg) between control and intervention groups.

based on initial hypertension severity. An instance of this is Kingue *et al*, where greater overall improvements (BP improved or BP at target) in participants with stage 3 hypertension was observed.[28] Owolabi *et al* also observed a significant reduction in BP for a subset of subjects with baseline BP >140.90 mm Hg but not an overall significant reduction for all participants.[41]

In terms of health knowledge improvement, in one study, CVD nurses reported that their own hypertension awareness and knowledge increased as well as that of the community, due to the ComHIP project.[47] Hacking *et al* found no statistical change in overall health knowledge, however, medication adherence was significantly higher due to intervention and self-reported behaviour change improvements.[34] Haricharan *et al* reported significant improvements in overall knowledge of healthy living and hypertension following exposure to informative SMS messaging.[35] PINGS resulted in significant medication adherence improvements,[39] and a trial of telemedicine for hypertension in Camaroon (TELEMED-CAM) saw significantly higher medical visit adherence in their intervention group.[28] Ola-Olorun *et al* reported positive perceptions towards an SMS-based medicine information exchange between patients and pharmacists, patients requested information on adverse medication effects.[29] Clinical documentation improved for all NCDs investigated by Kleczka *et al*, with a 21% improvement in hypertension documentation scoring.[36]

When perceptions towards the proposed technology were gathered, interventions were well received by patients[29 31 34 47] and health professionals.[29 37 44 47] Reported concerns included access to/stability of internet connection,[47] power availability,[37 44 47] cost,[44 47] increased workload,[38] understanding of SMS wording,[32] unfamiliarity with mobile technology or technology not being 'user-friendly'[34 44] and duplication in digital patient records.[49] Focus groups and interviews conducted by Adler *et al* indicated that health providers and policymakers identified major challenges in the use of a cloud-based health records system, which would require heavy reliance on outside resources.[47]

## DISCUSSION

Our systematic review of the literature found broad and diverse applications of KETs for tackling hypertension in SSA. The findings indicated that there is still relatively limited published research, particularly of controlled trials. All studies leveraged mobile phones for purposes of screening for hypertension, improving patient knowledge/treatment adherence or aiding non-physician healthcare workers in providing hypertension care. Other reviews targeted to SSA have focused on assessing either specific technological applications or different NCDs, and all noted a lack of published research.[18 20 50] Muiruri *et al*'s 2019 narrative literature review of telehealth interventions for hypertension in SSA[50] identified just eight studies, and in 2021, Osei *et al*[18] identified only 12 studies in a scoping review of mHealth for diagnosis or treatment of any disease in SSA. These authors also commented as we do on the paucity of studies of robust design, particularly those including control groups.[18 50]

Overall, our identified studies reported success in their outcomes, with overwhelmingly positive responses from participants towards the use of KETs. Consistent with our findings, other reviews comment on overall good acceptance of technologies by health workers[18 20] and SMS for health knowledge improvement and behaviour changes were identified as providing particularly promising positive results.[50] However, we found that very few studies were able to demonstrate statistically significant improvements over standard care, when evaluating objective measures such as BP reduction. This may indicate persistent difficulties in designing and implementing technology-based healthcare solutions in low-resource settings. Such difficulties were also evident from the quality analysis, for example, a frequent issue identified in the quality analysis for both randomised and non-randomised trials was a lack of complete outcome data, with authors describing difficulties with missing information and loss to follow-up as high as 50% in one study[35] and 58% in another,[48] an important consideration for future studies.

No publications using AI were identified in this study. Owoyemi *et al*[51] suggested that reasons for this may include limited available data, a lack of policy and legal framework, associated cost of uptake and inadequate infrastructure. Future research may explore predictive AI modelling either for screening and diagnostic tools and to identify and target the most promising areas for addressing patient lifestyle changes in SSA.

Many studies used KETs to facilitate task redistribution, which is a well-evidenced strategy to improving healthcare provision in areas with low numbers of qualified doctors/

specialists, for which the body of evidence relating to hypertension is growing.[52–57] Mobile technology provided decision support and record keeping tools aiming to empower non-physician workers in providing primary hypertension care. While perceptions and feedback from clinical staff and patients were overall positive, several key areas were consistently reported as major challenges for uptake of KETs. Fundamental issues in infrastructure are still a barrier to mobile technology for healthcare, evidenced by reported issues with internet, network and power coverage. In addition, several articles reported concerns around the ability of patients to use the technology and understand the information which was relayed, with calls for future research to investigate the feasibility and efficacy of audio visual rather than text communication.[34 42] Such findings have informed further development of the PINGS intervention, with upcoming phase III trials using a reductionist approach, removing a smartphone component and replacing with phone calls and audio and text messages.[58]

A notable finding was that few studies reported statistically significant benefits of KET-based interventions. Authors speculated that small sample sizes,[40] subject selection,[34 42] failed SMS delivery,[34] study design such that all patients received reminders,[33 45] free medication[33] or financial incentives[41] could have contributed to this, likely reflecting difficulty in retaining patients in care in SSA. It was also observed that interventions proved most effective among the highest risk groups, where it may be easiest to detect positive changes. Although not always found to be statistically significant, reductions in BP were observed, which, although modest, would, from a clinical perspective be anticipated to impact CVD development on a population level.[59] Our findings also indicated strategies using SMS to promote positive patient behaviour changes were highly successful.[34 39 47] It remains to be seen, however, whether self-reported behaviour changes translate into objective improvements in BP reduction.

### Strengths and limitations of this study

A major limitation of this systematic review was the heterogeneity of the included studies, which did not allow for quantitative synthesis of outcomes/results. Since this study also failed to identify any reports of use of AI, it is possible that extending the search beyond the scientific literature may have found cases where AI was intrinsic as part of manufactured technology already being used for healthcare in SSA. A strength of this study is that it is the first systematic review concerning use of KETs for hypertension healthcare in SSA, and in this way provides a comprehensive overview of the current state of the art and indicates gaps to be addressed in future research.

### CONCLUSION

Our study indicates that there is limited research on use of KETs for hypertension in SSA, particularly we did not identify any studies using AI. The study demonstrates that mHealth strategies provided positive impact on BP control, health knowledge and treatment adherence. Furthermore, stakeholder perceptions towards technology for hypertension prevention and management were positive. Therefore, further primary studies should be conducted, with an emphasis on objective measures such as BP reduction or BP control. It remains to be seen whether AI may also prove beneficial, such as through development of further diagnostic aids or boosting signals from cheap easily manufactured sensors.

**Contributors** KS, LP and FPPC designed the study. LP and FPPC coordinated and supervised the study. KS, BO, LP and FPPC designed the data collection and methodology. KS and BO screened records, extracted data and assessed quality. KS wrote the original draft. All authors critically revised and edited the manuscript. KS is the guarantor and accepts full responsibility for the work.

**Funding** KS is funded by the Medical Research Council Doctoral Training Partnership [grant number MR/N014294/1].

**Map disclaimer** The inclusion of any map (including the depiction of any boundaries therein), or of any geographic or locational reference, does not imply the expression of any opinion whatsoever on the part of BMJ concerning the legal status of any country, territory, jurisdiction or area or of its authorities. Any such expression remains solely that of the relevant source and is not endorsed by BMJ. Maps are provided without any warranty of any kind, either express or implied.

**Competing interests** None declared.

**Patient and public involvement** Patients and/or the public were not involved in the design, or conduct, or reporting, or dissemination plans of this research.

**Patient consent for publication** Not applicable.

**Ethics approval** Not applicable.

**Provenance and peer review** Not commissioned; externally peer reviewed.

**Data availability statement** All data relevant to the study are included in the article or uploaded as supplementary information.

**ORCID iDs**
Katy Stokes http://orcid.org/0000-0003-0766-6836
Francesco P Cappuccio http://orcid.org/0000-0002-7842-5493

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
