## [Reviewer comments · BMJ Open]

ARTICLE DETAILS

TITLE (PROVISIONAL)	Use of technology to prevent, detect, manage, and control hypertension in sub-Saharan Africa: a systematic review
AUTHORS	Stokes, Katherine; Oronti, Busola; Cappuccio, Francesco Paolo; Pecchia, Leandro

VERSION 1 – REVIEW

REVIEWER	Pellegrini, Giacomo Università degli Studi di Milano-Bicocca
REVIEW RETURNED	11-Nov-2021

GENERAL COMMENTS	The authors conducted a systematic review of the use of technologies to help control and treat hypertension in sub-Saharan Africa. As pointed out, there are not many studies available but this is the first systematic review of this topic for this geographic area and the methodology conducted by the authors in conducting the review was rigorous. My comments and suggestions follow: 1) The search was limited to studies published in English language? Explicit it. Although I guess it is difficult to do otherwise, given the geographical context of your research could it be a source of bias? If so, perhaps it is worth mentioning it as one of the limitations of the study.2) DISCUSSION, page 7 - line 54. The reference to logistic analysis is not clear to me. Why even among the supplementary materials you mention some articles that are not those selected by the systematic review? I believe that this is a passage that risks confusing the reader. I would suggest eliminating it by simply reporting the absence of studies on artificial intelligence3) DISCUSSION. Since your research focuses on a specific geographic area, it would be interesting if you cited and commented the results of other systematic reviews (if available or RCTs) in similar geographic contexts. What they concluded regarding technology and hypertension? Minor comments: - INTRODUCTION, page 4, line 26. It should be the first time the acronym NCD appears. Although known, it would be better to make it explicit.- FIGURE 2. I would recommend also annotated the name of the African state, along with the number of studies.- FIGURE 3. In this figure the only data is the number of studies for each type of technology. Why not just report the total in table 2.
---

	If you think the figure is useful, it is advisable to specify at least the frequency for each category or perhaps use another type of more immediate graph (e.g. bar chart). - For greater precision when using the supplementary materials reference replace the term "appendix" with the precise reference to the table or figure (e.g. supplementary table 1 and so on).
REVIEWER	Buabeng, Kwame Kwame Nkrumah University of Science and Technology
REVIEW RETURNED	13-Nov-2021
GENERAL COMMENTS	A very good review paper and well written. Weaknesses in some of the studies used in the systematic review was discussed as well as the limitations, including heterogeneity in the studies used for this systematic review. Authors may consider the removal of the word used in the abstract so the objective in the abstract could read [To identify and assess the use of technologies, including mobile health technology, internet of things devices and artificial intelligence in hypertension healthcare in sub-Saharan Africa (SSA)]

VERSION 1 – AUTHOR RESPONSE

Reviewer: 1
comments

The authors conducted a systematic review of the use of technologies to help control and treat hypertension in sub-Saharan Africa. As pointed out, there are not many studies available but this is the first systematic review of this topic for this geographic area and the methodology conducted by the authors in conducting the review was rigorous.

We thank the reviewer for their comments. We have addressed the following comments, and now feel the quality of the manuscript has improved.

2.1

The search was limited to studies published in English language? Explicit it. Although I guess it is difficult to do otherwise, given the geographical context of your research could it be a source of bias? If so, perhaps it is worth mentioning it as one of the limitations of the study.

Following the reviewer's comment, the fact that the search was limited to English language studies has been made explicit. Regarding exclusion of non-English papers, we understand this could have reduced the generalizability of the findings, despite being common in systematic reviews. Morrison et al. in "The effect of English-language restriction on systematic review-based meta-analyses: a systematic review of empirical studies" found no evidence of a systematic bias from the use of language restrictions in systematic review-based meta-analyses in conventional medicine. Moreover, we believe that the extent and effects of language bias may have diminished recently because of the shift towards publication of studies in English.

Specifically:

Page 2 Strengths and limitations of this study:

- 'The search was limited to studies published in English language.'

Methods (page 3 paragraph 1)

'We searched Embase, MEDLINE, Web of Science and Scopus electronic databases for studies published in English language only.'

2.2

DISCUSSION, page 7 - line 54. The reference to logistic analysis is not clear to me. Why even among the supplementary materials you mention some articles that are not those selected by the systematic review? I believe that this is a passage that risks confusing the reader. I would suggest eliminating it by simply reporting the absence of studies on artificial intelligence

We thank the reviewer for pointing out this possible area for confusion.

Logistic regression (LR) is a technique that can be used for building machine learning models, and therefore can be classifiable as an artificial intelligence method depending on the context in which it is employed. This is the rationale for including this technique in our search. In the studies we identified using LR, it was not employed as part of a machine learning system. The paragraph the reviewer refers to was written with intent to provide further information which, although out of scope, may be of interest to readers. We now appreciate this is likely to cause confusion and be detrimental to the manuscript, so have now removed this section. Further, this reduction below the word limit has allowed us to properly address the reviewers following comment on expanding the discussion.

Discussion paragraph 3 (page 9) now reads:

No publications using AI were identified in this study. Owoyemi et al.⁵¹ suggested that reasons for this may include: limited available data, a lack of policy and legal framework, associated cost of uptake and inadequate infrastructure. Future research may explore predictive AI modelling either for screening and diagnostic tools, and to identify and target the most promising areas for addressing patient lifestyle changes in SSA.

2.3

3) DISCUSSION. Since your research focuses on a specific geographic area, it would be interesting if you cited and commented the results of other systematic reviews (if available or RCTs) in similar geographic contexts. What they concluded regarding technology and hypertension?

We thank the reviewer for highlighting this and have now deepened the discussion relating to comparing and commenting on our results in the context of other reviews.

Discussion (page 8 paragraphs 1 and 2):

Our systematic review of the literature found broad and diverse applications of KETs for tackling hypertension in SSA. The findings indicated there is still relatively limited published research, particularly of controlled trials. All studies leveraged mobile phones for purposes of screening for hypertension, improving patient knowledge/treatment adherence or aiding non physician healthcare workers in providing hypertension care. Other reviews targeted to SSA have focused on assessing either specific technological applications or different NCDs and all noted a lack of published research.^{18 20 50} Muiruri et al.'s 2019 narrative literature review of telehealth interventions for hypertension in SSA⁵⁰ identified just eight studies and in 2021 Osei et al.¹⁸ identified only 12 studies in a scoping review of mobile health (mHealth) for diagnosis or treatment of any disease in SSA. These authors also commented as we do on the paucity of studies of robust design, particularly those including control groups.^{18 50}

Overall, our identified studies reported success in their outcomes, with overwhelmingly positive responses from participants towards the use of KETs. Consistent with our findings, other reviews comment on overall good acceptance of technologies by health workers^{18 20} and SMS for health knowledge improvement and behaviour changes were identified as providing particularly promising positive results.⁵⁰

INTRODUCTION, page 4, line 26. It should be the first time the acronym NCD appears. Although known, it would be better to make it explicit.

We thank the reviewer for noticing this discrepancy. The acronym NCD is included in line 1 of the introduction, where the term 'non-communicable disease' appears for the first time.

Introduction (line 1):

Cardiovascular disease (CVD) remains the most common cause of death due to non-communicable disease (NCD) worldwide, with 78% of deaths occurring in low- and middle-income countries (LMICs).

2.5

FIGURE 2. I would recommend also annotated the name of the African state, along with the number of studies.

We have now included the country names as annotations to the figure. In doing so, we realised one study was missed from the figure in error, this has now been corrected. We added a line in the figure legend to clarify that where several publications occurred from one study, the location is reported once. Table 2 column 'Study location' was modified slightly to highlight when a location was linked to a previous publication. We feel this makes it clearer for the reader.

Results (page 6 Figure 2 legend)

Countries are coloured based on the number of studies conducted (darker indicates more studies) and annotated with frequency (where a large study had several associated publications, the location is reported once).

2.6

FIGURE 3. In this figure the only data is the number of studies for each type of technology. Why not just report the total in table 2.

If you think the figure is useful, it is advisable to specify at least the frequency for each category or perhaps use another type of more immediate graph (e.g. bar chart).

We thank the reviewer for pointing this out. We have now reported the totals in table 2 and removed figure 3.

Results (page 6 paragraph 1):

Table 2 describes the different applications of technologies and their frequency of use.

Results (page 6 table 2 headings):

SMS (11), Smartphone/Tablet with App (8), Mobile/Smartphone/Tablet without App (6), IoT (3), Web-based Data Storage and Tools (7)

2.7

For greater precision when using the supplementary materials reference replace the term "appendix" with the precise reference to the table or figure (e.g. supplementary table 1 and so on).

We have addressed this and replaced any reference to 'appendix' with the precise reference to the supplementary, as suggested.

Specifically:

Methods (page 3 paragraph 2):

Search terms covered hypertension (e.g., "hypertension", "high blood pressure"), Artificial Intelligence (e.g., "AI", "machine learning"), mobile phones (e.g., "mobile phone*", "mobile"), internet of things (e.g., "internet of things", "iot"), point of healthcare cascade (e.g., "prevention", "screening") and countries of SSA (full strings in Supplementary S1).

Results (page 5 paragraph 2):

Quality was highest for qualitative and quantitative descriptive methods and relatively low for randomised or non-randomised studies (Supplementary Table S2).

Review: 2

Comments

A very good review paper and well written. Weaknesses in some of the studies used in the systematic review was discussed as well as the limitations, including heterogeneity in the studies used for this systematic review.

We thank the reviewer for their comments.

3.1

Authors may consider the removal of the word used in the abstract so the objective in the abstract could read [To identify and assess the use of technologies, including mobile health technology, internet of things devices and artificial intelligence in hypertension healthcare in sub-Saharan Africa (SSA)]

We thank the reviewer for noticing this, the redundant word has now been removed as suggested:

Abstract:

To identify and assess the use of technologies, including mobile health technology, internet of things devices and artificial intelligence used in hypertension healthcare in sub-Saharan Africa (SSA).

Error noted by authors

We noticed during revising of the manuscript that reference 32 appeared before reference 33. We have now corrected this error throughout the manuscript and in the reference list. We apologise for this discrepancy.

19th Jan additional corrections from email

1.

- Please provide a data availability statement in your main document. Kindly specify what unpublished data are available and where it can be accessed. If there are none then you can simply state "No additional data available".

Please note that the statement in the ScholarOne system and main document should be the same.

Data availability statement has been added, this is consistent with the statement in ScholarOne.

End of page 9:

Data sharing statement: All data relevant to the study are included in the article or uploaded as supplementary information. No additional data available.

2.

Research Ethics Approval must have separate sub-heading in main doc (word). Please note that this was mentioned under Methods. Below is the statement that was declared in Scholar One:

"Ethical approval was not required."

A separate sub-heading has been added in the main document.

Page 4:

Research Ethics Approval

Ethical approval was not required.

VERSION 2 – REVIEW

REVIEWER	Pellegrini, Giacomo Università degli Studi di Milano-Bicocca
REVIEW RETURNED	19-Jan-2022
GENERAL COMMENTS	The authors replied in a coherent and detailed point-by-point way to each annotation, modifying the paper where needed. I suggest that the paper is now ready to be accepted.